# Extracellular HSPs: The Potential Target for Human Disease Therapy

**DOI:** 10.3390/molecules27072361

**Published:** 2022-04-06

**Authors:** Dong-Yi Li, Shan Liang, Jun-Hao Wen, Ji-Xin Tang, Shou-Long Deng, Yi-Xun Liu

**Affiliations:** 1Key Laboratory of Prevention and Management of Chronic Kidney Disease of Zhanjiang, Institute of Nephrology, Affiliated Hospital of Guangdong Medical University, Zhanjiang 524001, China; lidongyi@gdmu.edu.cn (D.-Y.L.); liangshan@gdmu.edu.cn (S.L.); wenjunhan@gdmu.edu.cn (J.-H.W.); 2National Health Commission of China (NHC) Key Laboratory of Human Disease Comparative Medicine, Institute of Laboratory Animal Sciences, Chinese Academy of Medical Sciences and Comparative Medicine Center, Peking Union Medical College, Beijing 100021, China; 3State Key Laboratory of Stem Cell and Reproductive Biology, Institute of Zoology, Chinese Academy of Sciences, Beijing 100101, China

**Keywords:** heat shock proteins, extracellular HSPs, exosomes, inflammation, immune responses, molecular chaperones, unconventional protein secretion, cancers, neurodegenerative diseases

## Abstract

Heat shock proteins (HSPs) are highly conserved stress proteins known as molecular chaperones, which are considered to be cytoplasmic proteins with functions restricted to the intracellular compartment, such as the cytoplasm or cellular organelles. However, an increasing number of observations have shown that HSPs can also be released into the extracellular matrix and can play important roles in the modulation of inflammation and immune responses. Recent studies have demonstrated that extracellular HSPs (eHSPs) were involved in many human diseases, such as cancers, neurodegenerative diseases, and kidney diseases, which are all diseases that are closely linked to inflammation and immunity. In this review, we describe the types of eHSPs, discuss the mechanisms of eHSPs secretion, and then highlight their functions in the modulation of inflammation and immune responses. Finally, we take cancer as an example and discuss the possibility of targeting eHSPs for human disease therapy. A broader understanding of the function of eHSPs in development and progression of human disease is essential for developing new strategies to treat many human diseases that are critically related to inflammation and immunity.

## 1. Introduction

Heat shock proteins (HSPs), the conserved molecular chaperone proteins, were originally assumed to be stress-responsive proteins required for cell or organism survival after exposure to thermal stress. Shortly afterwards, it was found that HSPs could be induced by a wider variety of insults, such as inflammation, ischemia, and oxidative stress, in order to protect cells from further injury or death [1,2,3,4,5]. Although they had been named heat shock proteins or heat stress proteins, many of the HSPs are ubiquitously expressed in almost all types cells even under the physiological conditions, as they are essential for maintaining cellular homeostasis, such as protein homeostasis (proteostasis) [6,7,8,9].

According to their molecular weight and the characteristic domains, eukaryotic cellullar HSPs can be divided into several classes, such as Hsp60, Hsp70, and Hsp90 chaperones. Different classes of HSPs possess specific activities and functions, and sometimes they often work together to help native protein folding and maintain correct protein conformations, and therefore maintain their biological activities [10,11,12]. Moreover, HSPs can also re-fold denatured peptides, inhibit the aggregation of incorrectly folded proteins, and assist the defective or irreversibly misfolded proteins or peptides for degradation through the proteasomal and autophagic pathways [10,13]. HSPs often function as complexes which include other HSPs, co-chaperones, various accessory proteins, as well as ATPase activity modulators [14,15,16]. In 2009, clear nomenclature for HSPs was proposed and, currently, they are divided into six major and broadly conserved families, i.e., HSP100s, HSP90s, HSP70s, HSP60s, HSP40s and small HSPs [17].

For the last 20 years, HSPs have been considered to be typical intracellular proteins since the report of the heat shock response and characterization of the 70 kD heat shock proteins. However, growing evidence suggests that HSPs can be released into the extracellular space and blood and that they play essential roles in many human diseases through the modulation of inflammation and immune responses [18,19,20,21,22]. In this review, first, we describe the types of eHSPs, and then discuss the mechanisms of eHSP secretion. Next, we highlight the function of eHSPs in the modulation of inflammation and immune responses. Finally, we discuss the roles of eHSPs in many human diseases, such as cancers and neurodegenerative diseases, and the possibility of eHSPs as targets for developing strategies to treat many human diseases that are closely linked to inflammation and immunity.

## 2. HSPs Can Be Released into the Extracellular Milieu and Can Be Taken up by Other Cells

The evidence that HSPs can be released and can be taken up was first supplied by Tytell et al. in 1986, they reported that HSPs could be transferred from gila to axon [23]. Three years later, another team also reported that HSPs could be released by cultured rat embryo cells [24]. The release of these HSPs could not be inhibited by the inhibitors of the common secretory pathway, such as monensin or colchicine, but could be blocked by adding the lysine analogue aminoethyl cysteine, suggesting that HSPs were selectively released through an uncanonical mechanism of secretion [24]. At that time, these findings were regarded as potential artifacts and were neglected for many years, as these findings were against the common knowledge that HSPs were merely expressed intracellularly, since they lacked a classic secretion signal peptide in their sequence [25,26].

eHSPs attracted the attention of researchers again in the year 2000. In 2000, Srivastava et al. showed that necrotic cells but not apoptotic cells could release HSPs, such as gp96, calreticulin, hsp90, and hsp70, into the extracellular milieu, which could stimulate antigen-presenting cells (APC), such as macrophages and dendritic cells, through the nuclear factor kappa B (NF-kB) pathway to secrete several different cytokines [27]. Similarly, Calderwood et al. found that recombinant HSP70 could bind with high affinity to the monocyte surface, which could elicit a rapid monocyte intracellular calcium flux, causing activation of the NF-kB pathway and production of cytokines, such as the proinflammatory cytokines tumor necrosis factor alpha (TNF-α), interleukin-1 beta (IL-1β), and IL-6 [28]. Moreover, they also found that the binding of recombinant HSP70 with monocyte could activate two different signal transduction pathways: one dependent on CD14 and intracellular calcium, leading to the production of IL-1β, IL-6, and TNF-α; and the other only dependent on intracellular calcium, merely causing the production of TNF-α [28]. These studies suggested that necrotic cells could passively release HSPs into the extracellular milieu, and that these eHSPs could stimulate monocyte activation through the nuclear factor kappa B (NF-kB) pathway to produce cytokines, and therefore, could modulate the immune system.

However, the potential role of eHSPs, such as HSP70, in modulating the immune system was challenged by several studies [29,30]. These studies assumed that the effect of eHSPs on the immune system was probably due to the contamination of bacterial lipopolysaccharide or other bacterial products, such as the exogenous antigenic peptides [29,30]. Subsequently, several teams have provided reliable evidence showing that eHSP70 was actually able to activate the immune cells, such as macrophages and monocytes [31,32]. Nowadays, growing evidence has shown that HSPs could be released by various types of cells in several ways, passively or actively, and therefore, the scientific community has fully accepted this phenomenon and more attention has been given to the function of eHSPs in many human diseases [19,20,21,33,34].

## 3. The Mechanism of HSP Transportation from Intracellular to the Extracellular Milieu

Whether HSPs are released passively after cellular necrosis or exported through an active mechanism independent of cellular necrosis has been debated, since they have been detected in extracellular milieu or in plasma. Basu et al. assumed that the release of HSP70 into the extracellular milieu was the consequence of cell lysis caused by necrotic but not apoptotic cell death [27]. Therefore, they supposed that HSPs were passively released by the necrotic cells due to incomplete cell membranes. However, Hightower and Guidon showed that the release of HSP70 to the extracellular milieu was not dependent on cellular necrosis, but through a selective release mechanism [24], which was further confirmed by Hunter-Lavin et al. [35]. They found that the release of Hsp70 into blood or culture medium from peripheral blood mononuclear cells was not due to cell damage, but through a nonclassical pathway involving lysosomal lipid rafts [35]. Therefore, it is very likely that HSPs can be released from cells through both passive and active pathways.

The secretion of proteins through the classic ER-Golgi pathway needs a consensus peptide signal to guide their transmembrane transport. However, most of the HSPs do not contain the consensus peptide signal. Moreover, inhibition of the ER-Golgi pathway via the typical inhibitors, such as brefeldin A, did not inhibit the release of HSP70 from living rat embryo cells [24]. These observations suggest that HSPs are probably actively exported via the nonclassic or unconventional secretory pathway but not the classic ER-Golgi pathway in living cells [36,37,38,39,40].

Unconventional protein secretion (UPS) is conducted by a complex protein secretion system, which can transport proteins, without a signal peptide or a transmembrane domain, across the plasma membrane or can even enter the extracellular milieu by bypassing the ER-Golgi pathway [41,42,43,44]. There are no unifying mechanisms for the UPS to transport the leaderless proteins [41,42]. The leaderless cargos are secreted from the cytosol to the extracellular milieu through several mechanisms, such as pore-mediated translocation across the plasma membrane, ABC transporter-based secretion, and autophagosome/endosome-based secretion [40,41,43,45,46,47] (Figure 1). Although the specific mechanisms of eHSP secretion have not been fully elucidated, all we know for certain is that eHSPs are transported through different mechanisms and these mechanisms may coexist and act together for the secretion of the same eHSP in one cell.

Exosomes are nano-sized extracellular vesicles (40–100 nm in diameter) that are secreted by various cell types, contain proteins, nucleic acids, or other cargos, and play essential roles in the communication between different cells [48,49,50,51,52]. As the important type of UPS, exosomes are also involved in the secretion of eHSPs [53,54,55,56,57]. The main function of HSPs is to maintain cellular proteostasis, which needs the ATP to provide energy. However, there are not enough ATP in the extracellular milieu to supply the HSPs, therefore, eHSPs cannot exert their intracellullar function to maintain protein activity. Exosomes can transport HSPs from one cell to another, therefore, this may be a novel cellular stress signal transduction pathway that can functionally compensate for the imbalanced state of the heat shock response among different cells, and therefore, maintain the organismal proteostasis [56].

## 4. eHSPs in the Modulation of Inflammation and Immune Responses

As mentioned above, intracellular HSPs function as molecular chaperones to maintain cellular proteostasis and play a role in cytoprotection after stressful stimuli. However, eHSPs are unable to achieve such functions due to the limitation of the environment. It is now well established that eHSPs have roles in functions that are different from the well-understood intracellular molecular chaperone role [58]. The main function of eHSPs is to modulate the immune system by activating various immune cells, such as dendritic cells (DCs), macrophages, and monocytes [59,60] (Figure 2).

### 4.1. eHSPs Activate DCs

DCs are a special group of immune cells that function as a bond between innate and adaptive immunity, which can take up antigens via surface receptors and process these antigens and present them to both CD8+ and CD4+ T cells [61]. Additionally, the activated DCs can also upregulate many molecules, such as the costimulatory molecules, cytokines, and chemokines; therefore, DCs are essential components of both innate and adaptive immune responses [62].

Basu et al. showed that necrotic cells could release eHSPs, such as gp96 and hsp70, which could induce DCs to upregulate the expression of antigen-presenting and co-stimulatory molecules [27]. Moreover, they demonstrated that these eHSPs activated DCs via the highly conserved NF-kB pathway [27]. The activation of antigen-presenting cells (APC) by necrotic cell-released eHSPs may be an excellent mechanism for an organism to respond to tissue cell death caused by various internal and external stimuli. In addition, recombinant human HSP60 and human inducible HSP72 can also promote human DC maturation and activate DCs to secrete the proinflammatory cytokines [63,64]. Further study has shown that these HSPs could be specifically internalized by the CD14(-), Toll-like receptor 4(-) monocyte-derived DCs via the receptor-mediated endocytosis [65]. Released or recombinant eHSPs promoted DCs maturation and cytokines secretion, probably through the Toll-like receptor 4 dependent signaling pathway, as DCs of C3H/HeJ mice deficiency of a functional Toll-like receptor 4 had no response to HSP60 stimulation [66].

### 4.2. eHSPs Activate Macrophages and Modulate Their Polarization

As the representative of the first line of defense in innate immune responses, macrophages play an essential role in the regulation of host inflammation and immune responses by performing phagocytic activity, delivering the proinflammatory and anti-inflammatory cytokines, and shaping the tissue microenvironment [67,68,69]. Macrophages can be polarized to M1 or M2 phenotypes according to the microenvironmental stimuli. M1 macrophages produce the proinflammatory cytokines, such as IL-6, IL-12, and TNFα, whereas, the M2 macrophages release the anti-inflammatory cytokines, such as IL-10 and TGFβ.

Released or recombinant eHSPs can stimulate macrophages to secrete various cytokines, such as IL-1β, TNF-α, GM-CSF, and IL-12 [27,70,71]. For example, human HSP60 can be recognized by macrophages, and can cause the rapid release of TNF-α or nitric oxide [70]. Moreover, the proinflammatory macrophage response of hsp60 can synergize with IFN-γ, causing the macrophage to express the Th1-promoting cytokines, such as IL-12 and IL-15 [70]. Therefore, necrotic cell-released eHSPs may be a danger signal for the innate immune system that plays an important role in the chronic Th1-dependent tissue inflammation.

Tumor-associated macrophage (TAM), recruited and activated by cancer cells, appears in the advanced stages of cancer progression and shows the M2-like phenotype [72,73]. TAMs can promote cancer growth through providing an immunosuppressive microenvironment for cancer cells [74]. Various eHSPs, such as eHSP90α, eHSP70, eHSP110, and eHSP27, can stimulate macrophages to polarize toward TAM-like macrophages possessing immunosuppressive and proangiogenic phenotypes, and therefore, contribute to the continue progression of cancer [75,76,77,78,79,80].

### 4.3. eHSPs Function as a Cytokine to Activate Monocytes

Monocytes, the essential components of the inflammation and innate immune response, are the vanguard soldiers of the immune system [81]. Monocytes of the circulating system can infiltrate to the damaged or inflamed tissues, where they can further differentiate into either macrophages or dendritic cells [82]. Moreover, monocytes can bridge innate and adaptive immune responses, and change the tumor microenvironment via several mechanisms; therefore, they can induce immune tolerance and promote angiogenesis and tumor metastasis [83].

eHSPs, such as eHSP70, can also activate monocytes through binding with the plasma membrane of monocytes and eliciting a rapid intracellular calcium flux, causing the activation of the NF-κB pathway and upregulating the expression of proinflammatory cytokines, such as TNF-α, IL-1β, and IL-6 [28]. Mechanistically, eHSP70 activate monocytes, probably through two different signal transduction pathways, one pathway dependent on intracellular calcium and CD14, and the other pathway only dependent on intracellular calcium; the intracellular calcium and CD14 dependent pathway results in increased expression of IL-1β, IL-6, and TNF-α, whereas, the intracellular calcium dependent pathway causes an increase in the expression of TNF-α but not IL-1β or IL-6. Therefore, eHSPs can function as cytokines to stimulate monocytes to release various proinflammatory cytokines via activating the NF-κB pathway. Thus, HSPs have at least two roles, i.e., one role is an intracellular function as a chaperone, and the other role is an extracellular function as a cytokine.

In summary, eHSPs can activate various immune cells via binding with them through specific receptors such as CD14 or Toll-like receptor 4, and can promote them to release various cytokines. In addition, eHSPs can also promote APCs, and modulate the monocytes or macrophages inflammatory response.

## 5. eHSP and Human Diseases

Given the essential role of eHSPs in the modulation of inflammation and immune response, they have been reported to play essential roles in many human diseases, such as cancers, neurodegenerative diseases, and kidney diseases [20,33,84,85].

### 5.1. eHSPs and Cancers

In order to survive, the tumor cells need to resolve their internal stresses, such as dysfunction of proteostasis because of the high levels of protein synthesis and the presence of mutant proteins, and the hostile tumor microenvironment (TME) caused by stresses, such as hypoxia, nutrient deprivation and acidosis [86]. Therefore, HSPs are often highly expressed in many human cancers so as to reduce the higher intracellular stress and maintain proteostasis of cancer cells, therefore, eHSPs are believed to contribute to cancer transformation and progression [87,88,89].

It is now clear that various cancer cells can secrete eHSPs to the extracellular milieu or are associated with their plasma membranes through several pathways, which may contribute to the poor outcomes of cancer patients [90,91]. For example, colorectal cancer cells secrete eHSP110, which can affect macrophage polarization promoting them to a profile of protumor, anti-inflammatory; whereas, downregulation of eHSP110 can promote the macrophages to a profile of cytotoxic, proinflammatory [79,92,93]. Moreover, TLR4 has been assumed to be involved in macrophages polarization caused by eHSP110. Additionally, HSP110 high colorectal tumor biopsies were infiltrated with protumoral macrophages, whereas, those with HSP110 low biopsies were infiltrated with cytotoxic macrophages, indicating that macrophage polarization caused by eHSP110 was probably involved in the poor outcomes of patients with high HSP110 expression [79]. Therefore, eHSPs can promote tumor progression through promoting macrophage polarization to tolerogenic (M2-like) profile, and targeting eHSPs may be a good way to inhibit the progression of cancers [60,94,95,96].

### 5.2. eHSPs and Neurodegenerative Diseases

Neurodegenerative diseases are the most common diseases of the nervous system, characterized by progressively losing the neurons and consistent deposition of proteins, such as amyloid-β, prion protein, tau, and α-synuclein, in the brain and in peripheral organs. Currently, there are no efficient treatments for neurodegenerative diseases which cause a great burden to families and societies [97,98,99,100].

Microglia-mediated neuroinflammation is the one of the most obvious characteristics shared by many neurodegenerative diseases, such as Parkinson’s disease, Alzheimer’s disease (AD), and amyotrophic lateral sclerosis [99]. Growing evidence suggests that the activation of microglial cells (macrophages in nervous system) in the central nervous system is heterogeneous and can be divided into two opposite types: M1 phenotype and M2 phenotype [99,101,102]. Microglia can function as either cytotoxic or neuroprotective, according to the phenotypes activated [103]. Given the essential role of eHSPs in the modulation of inflammation and immunity, they have been considered to be involved in the progression of human neurodegenerative diseases [20,104,105,106].

It has been found that glial cells can release eHSPs (such as eHSP70), which can be taken up by neuronal cells, causing increased tolerance to neurological stress, and therefore, promoting motoneuron survival during the process of trophic factor deprivation [107,108]. Moreover, in mice studies, it has been shown that administration of recombinant HSP70 could improve the phenotype of various neurodegenerative diseases, such as Alzheimer’s disease and amyotrophic lateral sclerosis [109,110,111,112]. In Drosophila, overexpression of eHSPs by adding a signal peptide to Hsp70 could alleviate the phenotype of Alzheimer’s disease, manifested as suppression of Aβ42 neurotoxicity in adult eyes, reducing cell death, alleviating locomotor dysfunction, and extending lifespan [113]. Mechanistically, eHSP70 can bind to Aβ42 via its holdase domain, and can mask Aβ42 neurotoxicity through promoting the accumulation of nontoxic aggregates, which is independent of its ATPase activity. Therefore, eHSPs or extracellular vesicle-associated HSPs, may be a potential strategy for treating various human neurodegenerative diseases [20].

### 5.3. eHSPs and Kidney Diseases

Inflammation and immunity are critically involved in the progression of various kidney diseases, including acute kidney injury (AKI) and chronic kidney disease (CKD) [114,115,116,117]. eHSPs have been found in the urine of patients with kidney disease, and may be a good, noninvasive biomarker of kidney damage [118]. For example, urinary eHSP72 levels increased 3 days before an AKI diagnosis in critically ill patients, which was much more sensitive than any other tested biomarkers [118,119]. Moreover, in mice studies, it has been shown that urinary eHSP72 was also a sensitive marker to monitor therapeutic interventions and recovery rate of tubules after an acute insult [119]. A higher urine eHSP70/creatinine ratio was also found in type 1 diabetic children as compared with healthy children, which may reflect the degree of kidney damage during the early phases of disease [120]. In patients with end-stage renal disease on dialysis and patients with CKD stages 4 and 5, the excretion of eHSP70 in urine was significantly increased [121,122]. Therefore, eHSP levels in urine or serum may be a perfect, noninvasive indicator to examine the stages of kidney diseases, which may contribute to develop individualized treatment programs.

Currently, it is still unclear whether eHSPs are good or bad in kidney diseases [123]. An increased level of eHSPs in serum or urine from patients with kidney diseases may be a consequence of the injury rather than a cause, considering the important role of HSPs in maintaining intracellular protein homeostasis. The increased eHSPs in serum or urine may be caused by cell membrane rupture of damaged cells with high expression of HSPs. However, considering the important role of infiltration and activation of inflammatory cells in the kidney in the progression of renal diseases and the important roles of eHSPs in the activation of immune cells and inflammatory response, we have good reason to believe that eHSPs may play an important role in the progression of renal diseases [124,125,126]. Therefore, it is necessary to further investigate the specific function of eHSPs in kidney diseases, although this work is faced with many difficulties. Only with a deep understanding of their function will it be possible to develop drugs that target eHSPs to treat kidney diseases.

### 5.4. eHSPs May Be a Potential Target for Cancer

Targeting HSPs may be a promising strategy in cancer treatment, considering the important role of HSPs in shaping cancer cells’ ability to survive, adapt, and evolve [127,128]. However, despite the tremendous efforts that have been given to develop anticancer drugs by targeting HSPs, there is still no one HSP inhibitor, thus far, that has been approved by U.S. Food and Drug Administration (FDA) for treating cancer clinically [88]. These unfortunate failures are due to many reasons. For example, HSP inhibition may have serious side effects due to their bad effects on normal cell homeostasis. Additionally, another challenge for development of anticancer treatment based on HSPs is the lack of a comprehensive and true understanding of the expression and functions of HSPs in the context of internal disease [88]. Therefore, developing a highly effective anticancer treatment based on HSPs remains to be a significant challenge. Recent studies have shown that there were evolutionarily distinct functions between intracellular and eHSPs, and therefore, targeting eHSPs may be a new direction to develop effective anticancer strategies [18].

Indeed, strategies such as specifically targeting eHSP90 by cell-impermeable inhibitors (such as geldanamicyn beads, the geldanamicyn derivative DMAGN-oxide, or the ganetespib derivative STA-12–7191) [129,130,131] or monoclonal antibodies blocking eHSP90 [130,132,133,134,135,136] were able to inhibit cancer cell migration and metastasis. Xin et al. reported a novel engineered eHSP nanoinhibitor, i.e., the zinc-aspartic acid nanofibers, which had a specific binding ability to eHSP90 [137]. By binding to eHSP90, zinc-aspartic acid nanofibers could decrease the level of eHSP90 in a tumor microenvironment, and therefore, could inhibit cancer cell proliferation, migration, and invasion, without harm to normal cells. Thus, targeting eHSPs, especially eHSP90, may be a promising therapeutic strategy for the treatment of cancers.

## 6. Conclusions

Intracellular HSPs function as molecular chaperones to reduce cellular stresses and protect cells from death through maintaining cellular proteostasis, whereas eHSPs cannot function as molecular chaperones due to the lack of proper circumstances, such as the deficiency of ATP. Nevertheless, eHSPs still play essential roles in extracellular milieu. One of their important functions is that they can modulate the inflammation and immune responses, and therefore, play essential roles in diseases that are closely related to inflammation and immunity, such as cancers, neurodegenerative diseases, and kidney diseases. Here, we introduced the process of eHSPs and found several pathways of eHSPs secretion. Moreover, we highlighted the function of eHSPs in the modulation of inflammation and immune response. Additionally, we discussed the function of eHSPs in many human diseases, including cancers, neurodegenerative diseases, and kidney diseases. Although a lot of studies related to eHSPs have been conducted, many important questions remain. For example, it is not entirely clear how intracellular HSPs are secreted outside the cells or into the cell membrane. Under what circumstances do HSPs need to be secreted to extracellular milieu? What are the differences in extracellular function of different eHSPs? Most importantly, what roles do eHSPs play in many human diseases, and how can we use them to treat these diseases? Therefore, only through in-depth studies of these questions can we have the opportunity to develop efficient therapeutic treatments by targeting eHSPs to treat many human diseases.

## Figures and Tables

**Figure 1 molecules-27-02361-f001:**
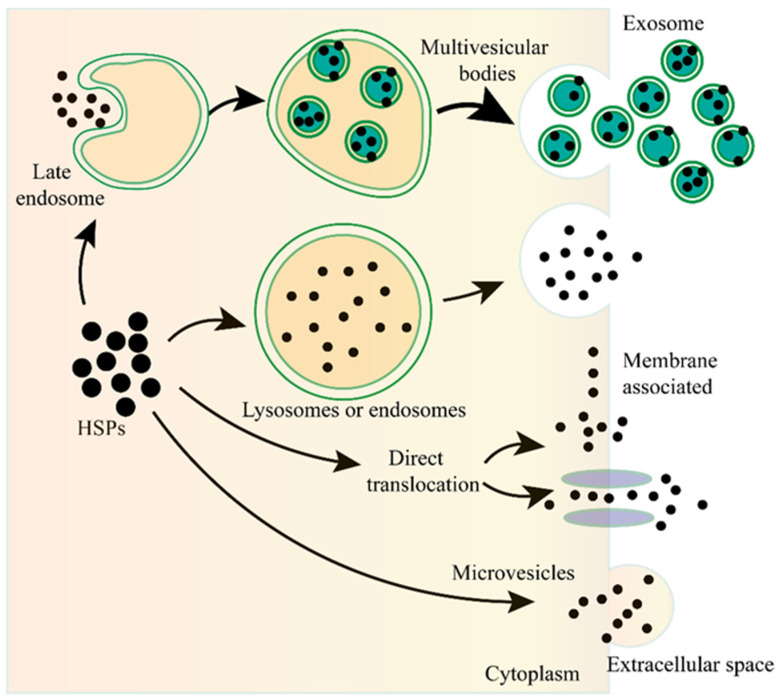
The unconventional secretion of Heat shock proteins (HSPs) outside the cell. After fusion of the lysosome or endosome with the plasma membrane, the HSPs can be released to the outside of the cell. HSPs captured from the cytoplasm can form vesicles, leading to the biogenesis of multivesicular bodies, and these internal vesicles are released outside the cell to become exosomes. HSPs can be directly translocated from the cytoplasm across the plasma membrane with or without the ATP binding cassette (ABC) transporter. HSPs can also be released into the extracellular space via microvesicles shed from the cell surface.

**Figure 2 molecules-27-02361-f002:**
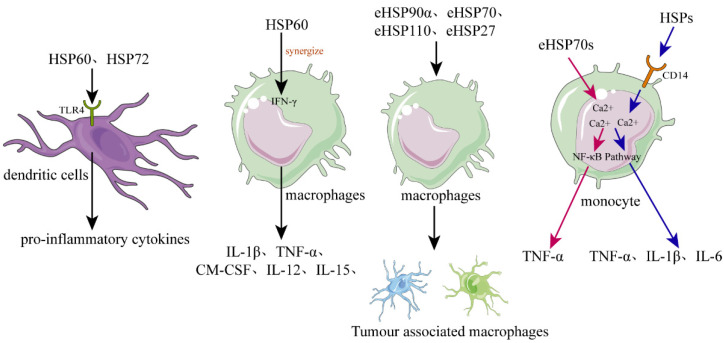
The role of HSPs in regulating inflammation and immune responses. In dendritic cells (DCs), recombinant human HSPs, such as HSP60 and HSP72, can promote their maturation and can activate them to secrete the proinflammatory cytokines. Released or recombinant eHSPs can stimulate macrophages to secrete various cytokines, such as IL-1β, TNF-α, GM-CSF, and IL-12, while hsp60 can synergize with IFN-γ. Various eHSPs, such as eHSP90α, eHSP70, eHSP110, and eHSP27, can stimulate the macrophages to polarize toward to TAM-like macrophages that possess immunosuppressive and proangiogenic phenotypes, promoting the progression of cancer. Regarding monocytes, they can be activated by eHSP70 probably through two different signal transduction pathways, the one pathway dependent on intracellular calcium and CD14, causing monocytes to increase the expression of IL-1β, IL-6, and TNF-α, and the other pathway only dependent on intracellular calcium resulting in increased expression of TNF-α but not IL-1β or IL-6.

## Data Availability

Not applicable.

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
