# Peer review of "Extracellular HSPs: The Potential Target for Human Disease Therapy"

_molecules, 2022, doi:10.3390/molecules27072361_

Round 1

Reviewer 1 Report

The Review « Extracellular HSPs: A Potential Target for Human Disease Therapy 3 by Dong-Yi Li  et al. is very interesting. 

The authors describe the various functions of extracellular Hsps in great detail, emphasizing their functions in modulating inflammation and immune responses.

However, the role of eHSPs in many human diseases is not fully presented in section 5.

For example, the role of HSP27, HSP60, HSP70, and HSP90 in cancer is insufficiently described. 

What is the role of HSPs in the pathophysiology and therapy of cardiovascular diseases? The authors wrote about this in the abstract but do not present the data in section 5.

I recommend expanding section 5 with new data and resubmitting the review after changes have been made.

The minor aspects:

Point 1

Please make the resolution clearer of Fig. 2.

Point 2

Please full expand all abbreviation in the special section.

Author Response

The Review « Extracellular HSPs: A Potential Target for Human Disease Therapy 3》by Dong-Yi Li  et al. is very interesting. 

The authors describe the various functions of extracellular Hsps in great detail, emphasizing their functions in modulating inflammation and immune responses.

However, the role of eHSPs in many human diseases is not fully presented in section 5.

For example, the role of HSP27, HSP60, HSP70, and HSP90 in cancer is insufficiently described. 

What is the role of HSPs in the pathophysiology and therapy of cardiovascular diseases? The authors wrote about this in the abstract but do not present the data in section 5.

I recommend expanding section 5 with new data and resubmitting the review after changes have been made.

Response: Thanks for your kind suggestions,which is valuable for improving the accuracy of the manuscript.  We have given more discussions in section 5 in the revised manuscript (Line 349-385). 

The minor aspects:

Point 1

Please make the resolution clearer of Fig. 2.

Response: Thanks for your kind suggestion. We have further optimized Figure 2 to make it look clearer. 

Point 2

Please full expand all abbreviation in the special section.

Response: Thank you very much for the excellent suggestion. We have full expanded all abbreviation in the special section (Line 34-43).

Reviewer 2 Report

Li et al, were responsible for an overview of heat shock proteins. Initially identified as proteins found at the cytoplasmic level, they have recently been found to play a predominant role as modulators of the inflammatory response and in immunity.  In fact, they appear to be involved in many human diseases, such as cancer, neurodegenerative and cardiovascular diseases, making them a hypothetical target for the treatment of these diseases.  The manuscript is well written, although there are a few parts that could be better explained.

I just have few minor remarks to make to the authors. 

In the context of the role of heat shock proteins in kidney disease, I suggest supplementing this part by referring to a recent publication (year 2021) by Junho et al in Cells, which is strongly focused on the role of these proteins in the pathophysiology of cardiac and renal changes. 

In addition, in the context of the role of these proteins in cancer, I would like to supplement this section by referring to a paper published in 2020 by Yun et al in Cells. 

Author Response

Li et al, were responsible for an overview of heat shock proteins. Initially identified as proteins found at the cytoplasmic level, they have recently been found to play a predominant role as modulators of the inflammatory response and in immunity.  In fact, they appear to be involved in many human diseases, such as cancer, neurodegenerative and cardiovascular diseases, making them a hypothetical target for the treatment of these diseases.  The manuscript is well written, although there are a few parts that could be better explained.

I just have few minor remarks to make to the authors. 

In the context of the role of heat shock proteins in kidney disease, I suggest supplementing this part by referring to a recent publication (year 2021) by Junho et al in Cells, which is strongly focused on the role of these proteins in the pathophysiology of cardiac and renal changes. 

Response: Thank you very much for the excellent suggestion. We have supplemented this part by referring to a recent publication (year 2021) by Junho et al in Cells (Line 349-361).

In addition, in the context of the role of these proteins in cancer, I would like to supplement this section by referring to a paper published in 2020 by Yun et al in Cells. 

Response: Thank you very much for the kind suggestion. We have supplemented this part by referring to a paper published in 2020 by Yun et al in Cells (Line 362-385).

Reviewer 3 Report

Dear authors,

Although the subject of the manuscript entitled "Extracellular HSPs: a potential target for human disease therapy" is of great relevance for the scientific community, the approach you chose to revise the relevant literature data is superficial. Moreover, there are two recently published review articles about eHSPs in cancer.

I suggest you focus on a specific disease or organ, i.e. neurodegenerative diseases, and write an in-depth revision of the literature, instead of a couple of paragraphs on the subject.

Furthermore, extensive editing of English language and style is required in order for the manuscript be considered for publication.

I regret to inform you that my decision is to reject the manuscript, and hope you will take the time to re-write the manuscript and submit it again.

Best Regards

Author Response

Dear authors,

Although the subject of the manuscript entitled "Extracellular HSPs: a potential target for human disease therapy" is of great relevance for the scientific community, the approach you chose to revise the relevant literature data is superficial. Moreover, there are two recently published review articles about eHSPs in cancer.

I suggest you focus on a specific disease or organ, i.e. neurodegenerative diseases, and write an in-depth revision of the literature, instead of a couple of paragraphs on the subject.

Response:Thank you for the constructive comments and suggestions. Our review focuses on the important role of eHSPs in regulating inflammation and immunity, and explores its impact on immune inflammation-related diseases and the possibility of targeting eHSPs to treat these diseases. Our purpose is not just to explore the role of eHSPs in a particular tissue or disease, as you said, similar work has been done by others.

Furthermore, extensive editing of English language and style is required in order for the manuscript be considered for publication.

Response: Thanks a lot for your comment. We have re-edited thoroughly the English language and style in the manuscript, and tracked changes to highlight the revisions. 

I regret to inform you that my decision is to reject the manuscript, and hope you will take the time to re-write the manuscript and submit it again.

Best Regards

Reviewer 4 Report

Li, Liang, Wen, et al. have written a review on the subject of extracellular heat shock proteins as targets in the treatment of diseases. The chapters on the release of HSPs and their roles in immune response mechanisms paint a very good picture of the current knowledge and are well referenced. The chapter on eHSPs and diseases is however quite short and could be more detailed. Notably, the paragraph on kidney disease. eHSPs here serve as biomarkers, but it is unclear from the manuscript if they have any biological role. An increase in eHSPs can be expected of any tissue that is damaged, the increase in eHSPs in urine from patients with kidney injury is therefore possibly more of a consequence of the injury than a cause, and eHSPs may not have any further role in this particular disease. Therefore, the authors should state if there is a specific significance of eHSPs in kidney disease other than simply serving as a biomarker.

The title is misleading as it mentions the targeting of eHSPs in diseases, but the text itself does not mention the targeting of eHSPs (other than a few short lines on their use in neurodegenerative diseases). In reading the title, I was expecting a chapter on the use of inhibitors of HSPs, antibodies or vaccines targeting eHSPs in cancer for example. And similarly if there are any HSP stimulating molecules used in neurodegenerative disorders. I understand that it would be difficult to distinguish the effect of modulators on intracellular and extracellular HSPs, nevertheless, if targeting HSPs is the subject, the authors must address this aspect. If the authors decide to add such a paragraph, I believe it would also be useful to add some information on the use of HSPs as adjuvants in vaccines, as these are extracellular, immune modulators and serve to protect against diseases. Also, if the authors decide to write such a paragraph, it would be of interest to state the possible problems of targeting eHSPs in diseases, notably since as the authors rightly note: eHSPs have a deleterious impact in a cancer context but are apparently positive in a neurodegenerative disorder context. If one targets eHSPs in cancer, does the patient risk of neurodegenerative disorder increase, and vice versa?

Finally, the authors should have their manuscript reviewed by a native English speaker. There are too many grammatical errors in the manuscript in its current form.

Author Response

Li, Liang, Wen, et al. have written a review on the subject of extracellular heat shock proteins as targets in the treatment of diseases. The chapters on the release of HSPs and their roles in immune response mechanisms paint a very good picture of the current knowledge and are well referenced. The chapter on eHSPs and diseases is however quite short and could be more detailed. Notably, the paragraph on kidney disease. eHSPs here serve as biomarkers, but it is unclear from the manuscript if they have any biological role. An increase in eHSPs can be expected of any tissue that is damaged, the increase in eHSPs in urine from patients with kidney injury is therefore possibly more of a consequence of the injury than a cause, and eHSPs may not have any further role in this particular disease. Therefore, the authors should state if there is a specific significance of eHSPs in kidney disease other than simply serving as a biomarker.

Response: Thanks for your kind suggestions,which is valuable for improving the accuracy of the manuscript.  We have given more discussions in section 5 in the revised manuscript (Line 349-361).

The title is misleading as it mentions the targeting of eHSPs in diseases, but the text itself does not mention the targeting of eHSPs (other than a few short lines on their use in neurodegenerative diseases). In reading the title, I was expecting a chapter on the use of inhibitors of HSPs, antibodies or vaccines targeting eHSPs in cancer for example. And similarly if there are any HSP stimulating molecules used in neurodegenerative disorders. I understand that it would be difficult to distinguish the effect of modulators on intracellular and extracellular HSPs, nevertheless, if targeting HSPs is the subject, the authors must address this aspect. If the authors decide to add such a paragraph, I believe it would also be useful to add some information on the use of HSPs as adjuvants in vaccines, as these are extracellular, immune modulators and serve to protect against diseases. Also, if the authors decide to write such a paragraph, it would be of interest to state the possible problems of targeting eHSPs in diseases, notably since as the authors rightly note: eHSPs have a deleterious impact in a cancer context but are apparently positive in a neurodegenerative disorder context. If one targets eHSPs in cancer, does the patient risk of neurodegenerative disorder increase, and vice versa?

Response: Thanks for your kind suggestions,which is valuable for improving the accuracy of the manuscript.  We have given more discussions about the targeting of eHSPs in diseases in section 5 in the revised manuscript (Line 349-385).

Finally, the authors should have their manuscript reviewed by a native English speaker. There are too many grammatical errors in the manuscript in its current form.

Response: Thanks a lot for your comment. We have tried our best to re-edited thoroughly the English language and style in the manuscript, and tracked changes to highlight the revisions. 

Round 2

Reviewer 1 Report

The manuscript has been significantly improved and now warrants publication in Molecules

Author Response

The manuscript has been significantly improved and now warrants publication in Molecules.

Response: Thank you very much for your recognition of our work. Thank you again for your valuable advice.

Reviewer 3 Report

Dear authors,

Although some corrections were indeed performed, extensive editing of English language and style is still required in order for the manuscript be considered for publication. A lot of phrases are still hard to understand due to grammar errors.

As for the previous issues I had, I understand your objectives with the review and I changed my decision regarding rejection of the manuscript. Therefore I advise the editor for the publication of the manuscript, but only after major revision of the English language.

Author Response

Dear authors,

Although some corrections were indeed performed, extensive editing of English language and style is still required in order for the manuscript be considered for publication. A lot of phrases are still hard to understand due to grammar errors.

As for the previous issues I had, I understand your objectives with the review and I changed my decision regarding rejection of the manuscript. Therefore I advise the editor for the publication of the manuscript, but only after major revision of the English language.

Response: Thanks a lot for your comment. We have tried our best to further improve the English language and style in the manuscript, and tracked changes to highlight the revisions. 

Reviewer 4 Report

The authors have added chapters and lines on the concerns I had over the lack of information on the targeting of eHSPs and on the function of eHSPs in kidney diseases. 

The english has improved, however there are still several mistakes and typos that should be corrected in order for the review to be accessible to readers.

Author Response

The authors have added chapters and lines on the concerns I had over the lack of information on the targeting of eHSPs and on the function of eHSPs in kidney diseases.

The english has improved, however there are still several mistakes and typos that should be corrected in order for the review to be accessible to readers.

Response: Thanks for your kind suggestions, which is valuable for improving the accuracy of the manuscript. We went ahead and made a lot of changes, hoping for a big improvement this time.